# Neonicotinoid Clothianidin reduces honey bee immune response and contributes to *Varroa* mite proliferation

Desiderato Annoscia [1,5], Gennaro Di Prisco[2,3,5], Andrea Becchimanzi[2], Emilio Caprio[2], Davide Frizzera [1], Alberto Linguadoca[2,4], Francesco Nazzi [1✉] & Francesco Pennacchio [2✉]

The neonicotinoid Clothianidin has a negative impact on NF-κB signaling and on immune responses controlled by this transcription factor, which can boost the proliferation of honey bee parasites and pathogens. This effect has been well documented for the replication of deformed wing virus (DWV) induced by Clothianidin in honey bees bearing an asymptomatic infection. Here, we conduct infestation experiments of treated bees to show that the immune-suppression exerted by Clothianidin is associated with an enhanced fertility of the parasitic mite *Varroa destructor*, as a possible consequence of a higher feeding efficiency. A conceptual model is proposed to describe the synergistic interactions among different stress agents acting on honey bees.

[1] Dipartimento di Scienze AgroAlimentari, Ambientali e Animali, Università degli Studi di Udine, Udine, Italy. [2] Dipartimento di Agraria, Laboratorio di Entomologia "E. Tremblay", Università degli Studi di Napoli "Federico II", Portici, Napoli, Italy. [3] Present address: CREA, Consiglio per la Ricerca in Agricoltura e l'Analisi dell'Economia Agraria, Centro di Ricerca Agricoltura e Ambiente, Bologna, Italy. [4] Present address: Department of Biological Sciences, Royal Holloway, University of London, London, UK. [5] These authors contributed equally: Desiderato Annoscia, Gennaro Di Prisco. ✉email: francesco.nazzi@uniud.it; f.pennacchio@unina.it

Neonicotinoids have entered the pesticide market since 1990, becoming rapidly the most widely used insecticide molecules worldwide[1,2]. However, in the last decade, a number of sub-lethal effects on ecosystem service providers (i.e., biocontrol agents and pollinators) have been reported[3]. Among these sub-lethal effects, special attention has been devoted to the negative impact of neonicotinoids on honey bee immunity and health, showing that Clothianidin impairs NF-κB signaling and the downstream antiviral immune barriers, promoting intense replication of deformed wing virus (DWV) in honey bees bearing an asymptomatic infection[4].

However, neonicotinoids are only one of the many stressors impairing honey bee health and survival[5–7]. Among these, a preeminent role is played by the mite *Varroa destructor*, the most important ectoparasite of honey bees, which is a major problem for honey bee colonies in the Northern hemisphere[8,9]; this is largely related to its activity as a vector and activator of DWV[10,11], with which the mite has established a symbiotic association that exacerbates their respective impacts on honey bee health[12].

Recently, a significant increase of mite infestation in colonies adjacent to fields exposed to neonicotinoid treatments was reported[13,14], suggesting a possible link between these pesticides and mite population abundance[15]. A synergistic negative effect of the association between *Varroa* and neonicotinoid insecticides on honey bees has also been reported[16–19]. This interaction has been largely overlooked so far, in spite of its potential remarkable importance, since any factor exacerbating the effects of mite infestation can have dramatic consequences on the survival of honey bee colonies[20].

A model on how different stress factors concur in the negative modulation of honey bee immune-competence and the capacity to contain pathogens and parasites has been proposed[6,7,20]. Based on that model, we predicted that the negative effects of *Varroa*–DWV symbiotic association[12] can be exacerbated by any immune-modulating stressor triggering viral replication and/or facilitating *Varroa* feeding, enhancing so its fitness. In particular, the exposure of honey bees to neonicotinoids can, in theory, enhance *Varroa* proliferation as a consequence of the immune-suppressive effect of these insecticides[4,21], which can be further amplified by the induced viral replication[12].

To test this hypothesis, we teased apart the direct immuno-suppressive effect of Clothianidin by evaluating the alteration of the immune response induced by this neonicotinoid insecticide in honey bee larvae collected early in the season, bearing null or very low titers of DWV, and measured the resulting impact on the fitness of *Varroa* mites feeding on them. Indeed, a very low DWV infection pressure, which contributes to the reduction of honey bee immune-competence[4,12,20], is necessary to provide direct circumstantial evidence required to corroborate the invoked promoting effect of neonicotinoids on *Varroa* mite proliferation[13–15]. This hypothesis, if true, would shed light on the delicate issue of how the environmental contamination by xenobiotics, in particular pesticides, can affect the complex network of interactions existing in nature. Moreover, this new finding would add a further layer of complexity in a very much needed holistic view of environmental risk assessment strategy for pesticides[22].

Here we assess the impact of Clothianidin on the immune response of honey bees and the reproductive activity of *Varroa* mites feeding on them. We show that this neonicotinoid insecticide has a negative effect on honey bee immune-competence and wound healing, which is associated with an enhanced fertility of the parasitic mite, as a possible consequence of its higher feeding efficiency. Our results indicate that immune disrupters as Clothianidin can influence in multiple ways the intricate network of interactions among stress agents that have a synergistic impact on honey bee health.

## Results

In order to test if Clothianidin affects the immune response of the honey bee stage on which *Varroa* feeding and reproduction take place, we assessed the melanization and encapsulation of a nylon thread implanted in the body cavity of L5 honey bee larvae, treated with different insecticide doses. The rationale behind this approach is that the host immune response can interfere with food uptake and use by ectoparasitic arthropods, which, to counteract this problem, adopt a wealth of immunosuppressive strategies targeting both the humoral and cellular components[23]. The neonicotinoid Clothianidin has a negative effect on the activation of NF-κB[4], a transcription factor which upon immune challenge gets rid of the inhibitor IκB, enters the nucleus and activates genes regulating a number of humoral and cellular defense reactions in insects[24–26]. Therefore, in principle, this immunosuppressive insecticide could be able to enhance the efficiency of host nutritional exploitation by the feeding mite. This is apparently a very important functional constraint, which, indeed, may have driven the *Varroa*–DWV co-evolution process, leading to a tight symbiosis characterized by a concurrent viral-induced host immunosuppression and enhanced mite's fitness[12].

Both melanization and encapsulation appeared to be significantly reduced by Clothianidin, used at the concentrations of 0.01 and 0.05 ppm (Mann–Whitney $U$ test: $n_1 = 15$, $n_2 = 15$, $U = 0$, $P < 0.001$ for both melanization and encapsulation; Fig. 1a, b), which can be regarded as realistic field doses[27–30]. Moreover, parallel studies on adult honey bees showed that the observed effects are not stage specific and are clearly dose-dependent within a broader range of sub-lethal doses (melanization: Kruskal–Wallis, adj. $H = 70.17$, df = 4, $P < 0.001$; encapsulation: Kruskal–Wallis, adj. $H = 68.34$, df = 4, $P < 0.001$; clotting: Mann–Whitney $U$ test: $n_1 = 15$, $n_2 = 15$, $U = 4$, $P < 0.001$; Fig. 1d, e and Supplementary Fig. 1a).

This result is reasonably predictable considering the negative effect of Clothianidin on the activation of NF-κB[4] and the central role of this transcription factor in the activation of the molecular pathways controlling the immune response in honey bees, like in other insects[24–26]. To further corroborate this hypothesis, we focused on the expression profile, as affected by Clothianidin exposure, of a honey bee immune gene (*Amel\102*) involved in the melanization and encapsulation of foreign invaders, which is under NF-κB transcriptional control[12]. Indeed, the expression of *Amel\102* was significantly reduced in honey bee larvae treated with 0.01 ppm of Clothianidin, 72 h after treatment (Mann–Whitney $U$ test: $n_1 = 15$, $n_2 = 15$, $U = 58$, $P = 0.012$, Fig. 2a), while *Dorsal 1A* (a member of the NF-κB family) showed only a reduced trend of transcription rate (Mann–Whitney $U$ test: $n_1 = 15$, $n_2 = 15$, $U = 74$, $P = 0.055$, Fig. 2b). This is consistent with the fact that Clothianidin enhances the transcription of a gene encoding a negative modulator (*Amel\LRR*) of NF-κB activation, which results in the transcriptional downregulation of *Amel\102* as well as of other downstream genes encoding antimicrobial peptides[4,12].

It has been shown that DWV infection could have a negative effect on NF-κB activation by enhancing the transcription of *Amel\LRR* gene[12]. Therefore, to verify whether the disruption of the honey bee immune response observed above was partly affected by the viral replication, we used qRT-PCR to assess the DWV load in honey bee larvae treated with two doses of Clothianidin. We found that the treatment with Clothianidin caused a small but significant increase of DWV titer in the experimental honey bee larvae, which was similar for both doses used (Mann–Whitney $U$ test: $n_1 = 15$, $n_2 = 15$, $U = 0$, $P < 0.001$; Fig. 1c); the same trend was also observed in adult bees (Mann–Whitney $U$ test: $n_1 = 15$, $n_2 = 15$, $U = 4$, $P < 0.001$; Fig. 1f and Supplementary Fig. 1b). However, both the basal

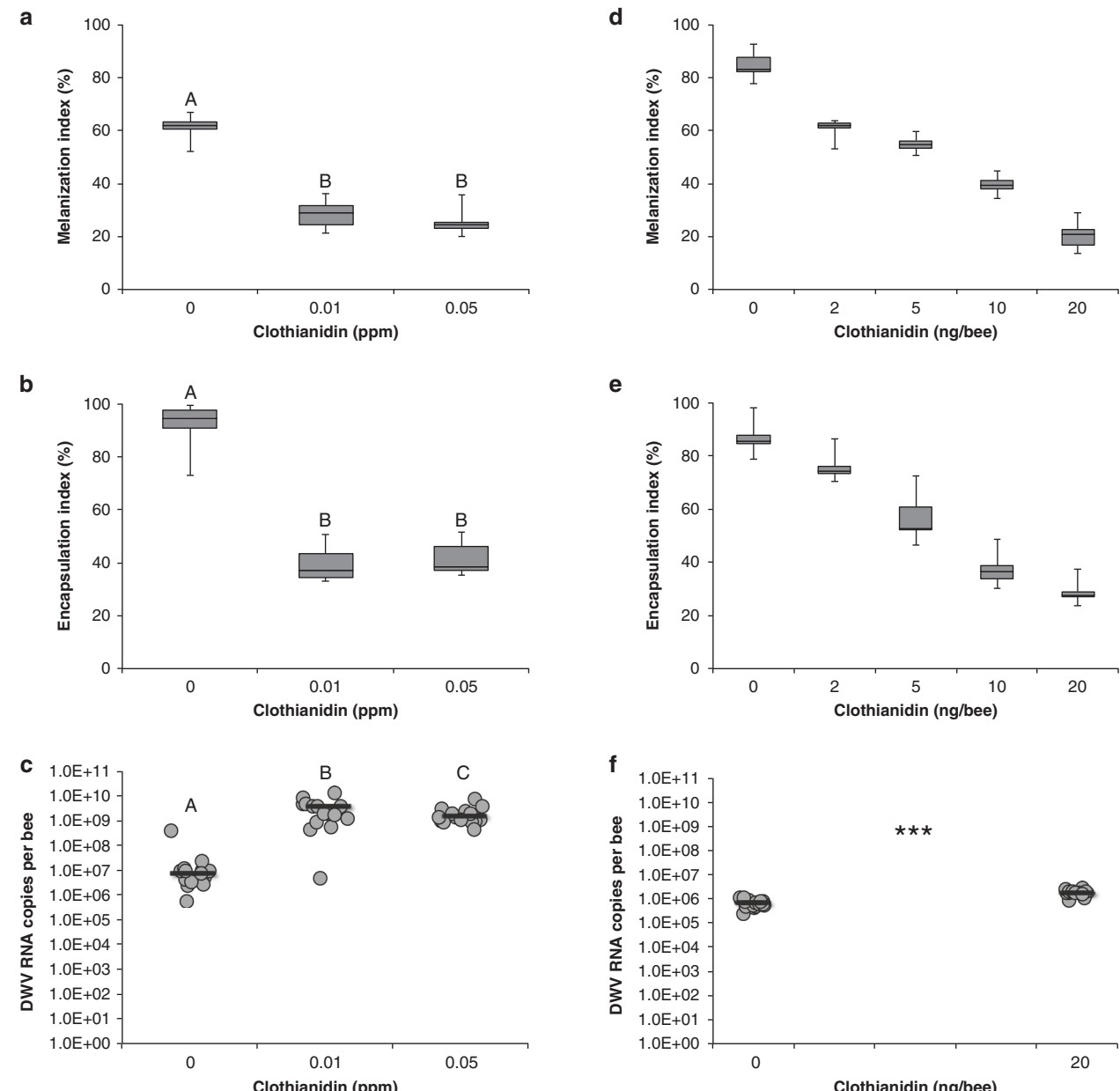

**Fig. 1 Effect of Clothianidin treatment on encapsulation and melanization of a nylon thread implanted in the body cavity of larvae and adults of honey bee.** For each experimental condition, 3 replicates of 5 honey bees each were considered. Range (from minimum to maximum value), 1st and 3rd quartile and median are reported in the box-plots; the horizontal bars in the scatter-jittered plots represent the sample average; different letters denote significant differences. **a** Melanization in mature bee larvae (0 vs 0.01 ppm and 0 vs 0.05 ppm: Mann–Whitney $U$ test: $n_1 = 15$, $n_2 = 15$, $U = 0$, adjusted, one tailed $P < 0.001$). **b** Encapsulation in mature bee larvae (0 vs 0.01 ppm and 0 vs 0.05 ppm: Mann–Whitney $U$ test: $n_1 = 15$, $n_2 = 15$, $U = 0$, adjusted, one tailed $P < 0.001$). **c** DWV RNA copies recorded in experimental honey bee larvae (0 vs 0.01 ppm and 0 vs 0.05 ppm: Mann Whitney $U$ test: $n_1 = 15$, $n_2 = 15$, $U = 0$, adjusted, one tailed $P < 0.001$; 0.01 vs 0.05 ppm: Mann–Whitney $U$ test: $n_1 = 15$, $n_2 = 15$, $U = 60$, adjusted, one tailed $P = 0.044$). **d** Melanization in adult bees (Kruskal–Wallis: adj $H = 70.17$, df = 4, $P < 0.001$). **e** Encapsulation in adult bees (Kruskal–Wallis: adj $H = 68.34$, df = 4, $P < 0.001$). **f** DWV titer recorded in adult bees used for the implantation experiment (Mann–Whitney $U$ test: $n_1 = 15$, $n_2 = 15$, $U = 4$, one tailed $P < 0.001$).

($10^6$) and insecticide induced ($10^9$) levels of viral genome copies were well below the symptomatic threshold[10] and the levels (e.g., $10^{15}$–$10^{18}$) found associated with an evident immune depression[4,12,20]. Then, the observed reduction of immune competence in the experimental larvae exposed to Clothianidin treatment is only very limitedly affected by DWV, if any.

The reported negative effect of Clothianidin both on humoral and cellular immune responses in honey bee larvae could have, as said, a significant impact on *Varroa* feeding, since the trophic activity of this parasite takes place on early honey bee pupae,

through a feeding hole which must remain pervious over time to allow efficient food uptake by the feeding mites[31,32]. Therefore, Clothianidin, by reducing the NF-κB mediated immune reaction in response to the feeding wound, would favor the trophic activity of the parasite.

To corroborate this hypothesis, we assessed the reproductive capacity of *Varroa* mites feeding on honey bee larvae exposed to Clothianidin and subjected to controlled infestation in vitro with a single mite. The adults emerging from the experimental honey bee larvae showed a low but significant enhancement of DWV

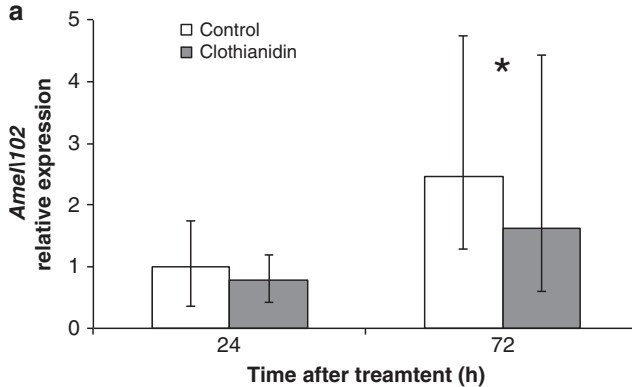

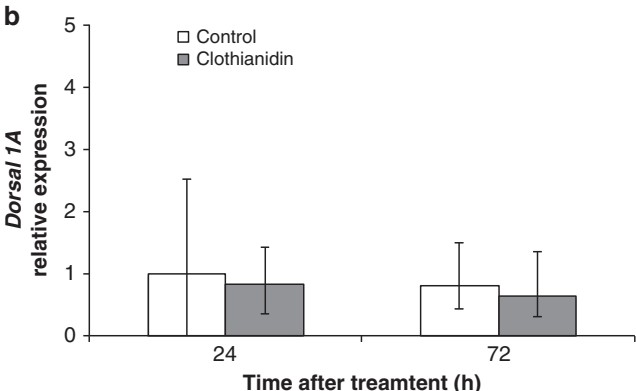

**Fig. 2 Relative expression over time of two genes involved in humoral and cellular immune response of honey bee larvae.** At each time point, 15 biologically independent larvae per treatment were sampled. The asterisk denotes a significant difference (Mann–Whitney $U$ test: $n_1 = 15$, $n_2 = 15$, $U = 58$, one tailed $P = 0.012$) for the mean values recorded at a specific time point; error bars represent the standard deviation of the mean, calculated according to the $\Delta\Delta Ct$ method. **a** Relative expression over time of $Amel\backslash 102$. **b** Relative expression over time of $Dorsal\ 1A$.

titer after 24 h exposure to Clothianidin (Mann–Whitney $U$ test: $n_1 = 28$, $n_2 = 27$, $U = 140$, $P < 0.001$; Fig. 3a). However, it is worth noting that this application of Clothianidin determined only a very limited increase of the low $(10^6)$ basal DWV titer, compared to the steep increase observed on honey bees with a much higher starting level of infection $(10^8–10^{11})^{12}$. Based on the previous data[20], we can assume that the immune-suppression mediated by the viral component is reasonably negligible under these experimental conditions (i.e. early in the season, with honey bee larvae bearing low levels of DWV infection).

Mite fertility (i.e., proportion of reproducing females out of total female mites used in the experiment[12]) on bee pupae treated with 0.01 ppm of Clothianidin at the larval stage was significantly higher than on control bees (Mantel–Haenszel test: df = 2, M–H Chi-2 = 3.970, $P = 0.046$; Fig. 3b) by 23%, on average; on the contrary, fecundity (i.e., number of offspring per reproducing female) seemed not to be affected (Mann–Whitney $U$ test: $n_1 = 78$, $n_2 = 68$, $U = 2487$, n.s.; Supplementary Fig. 2).

The observed increase of fertility in mites parasitizing Clothianidin treated bee larvae is possibly due to the fact that their feeding activity is facilitated by the induced immune-suppression of the host. Indeed, as hypothesized above, the reduced capacity to mount both a cellular and a humoral immune response, which would interfere with food uptake and use, can largely account for this result. This is further corroborated by the fact that the immune-suppression induced by $Varroa$ vectored DWV similarly enhances mite's fitness[12].

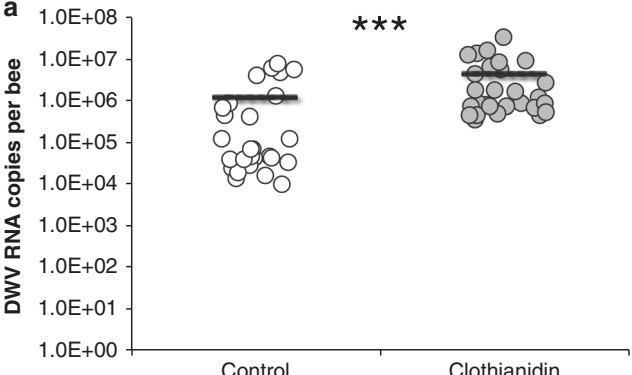

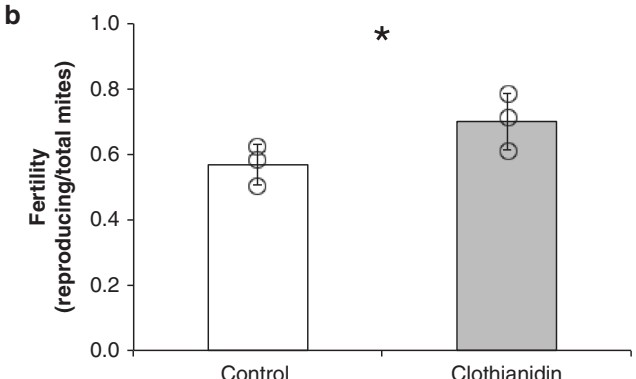

**Fig. 3 Clothianidin effect on DWV infection level and $Varroa$ reproduction.** Asterisks denote significantly different mean values (one asterisk: $P < 0.05$; three asterisks: $P < 0.001$). **a** DWV infection level in honey bees eclosed from larvae reared in vitro on a diet with or without Clothianidin (the experiment was run in triplicate, for a total of 28 and 27 individuals for Clothianidin treated and untreated controls, respectively; the horizontal bars represent the sample average; Mann–Whitney $U$ test: $n_1 = 28$, $n_2 = 27$, $U = 140$, one tailed $P < 0.001$). **b** Fertility of $Varroa$ mites on honey bee larvae treated with Clothianidin or on untreated controls (the experiment was run in triplicate, for a total of 111 and 120 individuals for Clothianidin treated and untreated controls, respectively; the proportion of reproducing mites in each replicate along with the average fertility and relative standard deviation are reported; Mantel–Haenszel test: df = 2, M–H Chi-2 = 3.970, $P = 0.046$).

This is a very interesting novel acquisition that sheds light on the possible causes of the unexpected proliferation of $Varroa$ mite in bee colonies exposed to neonicotinoid insecticides[13,14]. In fact, the doses used in our study are comparable with those found in pollen and nectar collected by bees maintained nearby treated crops[27–30], and thus reasonably expected in jelly fed to mature bee larvae.

To further test if the enhancement of $Varroa$ reproduction caused by the contamination of bee larvae with Clothianidin, as reported in this study, is compatible with the effects on mite infestation previously observed under field conditions, we developed a simple model of $Varroa$ population to simulate the dynamics of mite infestation in presence or absence of Clothianidin. Briefly, in our discrete time model, the mite population is calculated on a daily basis, using standard parameters as derived from the literature and corrected to include the observed effect of Clothianidin on mite reproduction.

We found that mite infestation in hives contaminated with Clothianidin could reach a level that is 1.4–2.0 times higher than

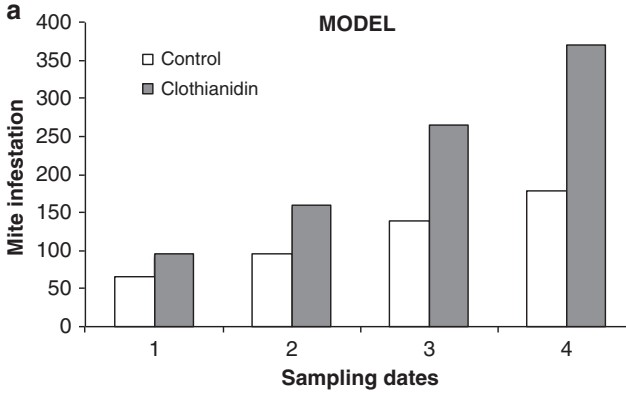

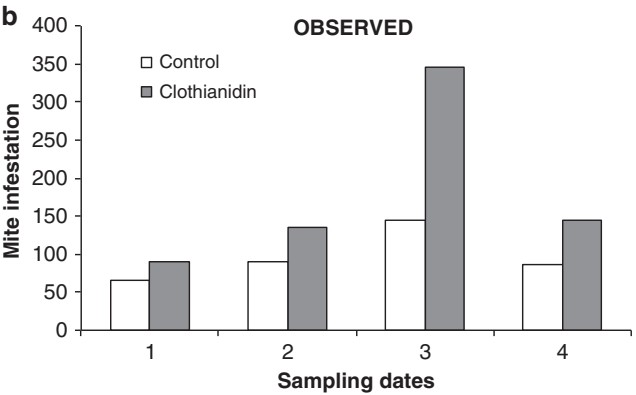

**Fig. 4 Comparison between predicted and observed mite infestation of hives at different sampling times during the season. a** Mite infestation as predicted using the discrete time model we developed and the reproduction data of *Varroa* mites as affected by Clothianidin exposure. **b** Mite infestation observed in hives close to fields treated with neonicotinoids[13].

that observed in uncontaminated hives, according to the season (Fig. 4a). This result matches quite well the observations carried out under field conditions[13], where a 1.4–2.4 higher mite infestation was observed in hives placed near corn fields planted with neonicotinoid-coated seeds (Fig. 4b).

## Discussion

This study further contributes to the elucidation of the complex network of interactions among different stress agents that have a negative impact on honey bee immune-competence and health (Fig. 5). In social insects the immune control of parasites and pathogens is of central importance[33,34]; the stability of this unique microcosm, based upon diversity and tolerance, relies on sophisticated and finely tuned mechanisms. The general concept emerging from this study is that the balance of such complex ecological communities is exposed to a wealth of risks generated by the unpredictable effects of different environmental stress agents that may disrupt the immune balance and the energy flow through the system. The key-point is that many of these stressors can be minor per se, but can become a problem when their synergistic interaction generates self-boosted loops of parasite/pathogen proliferation, as described in the proposed stress diagram (Fig. 5) and, more generally, in the immune model we proposed for interpreting the mechanistic basis of health decline and eventual losses of honey bee colonies[6,7,20]. The impact of pesticides on insect immunity can be quite relevant[35], especially if interpreted in the logical framework we propose. Indeed, a better knowledge about the underlying interactions among stressors could help to interpret the contrasting results obtained so far under field conditions, exposing honey bees to

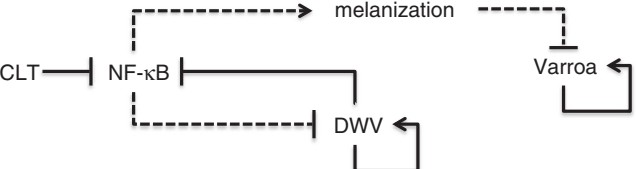

**Fig. 5 The network of the observed interactions among different stress agents.** Clothianidin (CLT) contamination decreases NF-κB activation, impairing the immune response and enhancing mite's fitness, as a possible consequence of a higher feeding efficiency. The decrease in NF-κB activation relaxes immune control on DWV, which, above a certain threshold, impacts NF-κB, reinforcing the negative effect on immune response. Arrows indicate positive (i.e., stimulation or upregulation) interactions; bar-headed lines mark negative interactions (i.e., inhibition or downregulation); dashed lines indicate that the effect can be impaired by Clothianidin treatment.

neonicotinoid insecticides in presence of different parasite and pathogens loads[36] (and citations therein).

The important effect of pesticides on insect immunity should receive more attention in the definition of novel protocols for risk assessment and in the study on how pesticides can interfere with the complex ecological network existing in the hive microcosm.

## Methods

**Impact of Clothianidin on melanization and clotting.** *Insects*: honey bees used in this study were from *Apis mellifera ligustica* colonies, maintained in the experimental apiary of the University of Napoli "Federico II", Department of Agricultural Sciences. Larvae and newly emerged bees used in all the experiments were obtained from brood frames taken from the experimental hives and kept in an incubator at 34 °C, 80% relative humidity for 12 h.

*Implantation experiment*: 3rd instar larvae were first fed with 0.05, 0.01 ppm and no Clothianidin, while adults were treated with 20.0, 10.0, 5.0, 2.0 ng/bee and no Clothianidin, as already published[4] (5 individuals for each treatment for both larvae and adults). In order to evaluate the encapsulation and melanization index[12] a piece of transparent, nylon fluorocarbon coated fishing line (Ø = 0.08 mm; Asso Fishing Line), sterilized under UV light for 24 h, was inserted into the hemocelic cavity on 4th body segment of 5th instar larvae and into the haemocoelic cavity of adults through the membrane between the 3rd and 4th abdominal tergite. After 24 h, the implants were removed and subjected to image analysis, using GIMP version 2.8 (GNU Image Manipulation Program; www.gimp.org). In adult bees the clotting index was also analyzed by evaluating, after 24 h, the healing of a wound generated by piercing the honeybee integument inter-membrane between the 3rd and 4th abdominal tergite, using a sterile entomological needle. The rest of body was immediately stored at −80 °C for the subsequent molecular analysis. The experiment was repeated 3 times.

*Immune genes expression and DWV quantification*: in order to assess the relative expression of *Amel\102* and *Dorsal 1A* as affected by Clothianidin treatment, two groups of 4th instar larvae (*n* = 100 per group) received 0.01 ppm of a Clothianidin-treated diet or a clean diet, respectively, as detailed below. After 24 and 72 h from feeding, 15 larvae for each experimental group were sampled and stored at −80 °C for subsequent analysis.

RNA extraction, DWV quantification and relative gene expression data analysis were performed according to already published protocols[12]. Briefly, total RNA was isolated from individual honey bees using TRIzol reagent (Thermo Fisher Scientific, Waltham, MA, USA), according to the manufacturer's instructions. The quantity and the quality of total RNA were assessed using Varioskan Flash spectrophotometer (Thermo Fisher Scientific).

Differential relative expression of *Amel\102* and *Dorsal 1A* was measured by one-step qRT-PCR, using the Power SYBR Green RNA-to-Ct 1-Step Kit (Applied Biosystems, Carlsbad, CA, USA), according to the manufacturer's instructions. Each reaction was prepared in 20 μL and contained 10 μL qRT-PCR mix 2X, 100 nM of forward and reverse primers, 0.16 μL of 125X RT enzyme mix, DEPC treated water and 50 ng of total RNA. All samples were analyzed in duplicate on a Step One Real Time PCR System (Applied Biosystems). Two reference genes, *β-actin* and *rps5*, were used as endogenous control for RNA loading. Relative gene expression data were analyzed using the ΔΔCt method.

The quantification of DWV genome copies was performed using the Power SYBR Green RNA-to-Ct 1-Step Kit (Applied Biosystems) as described above. Titers of DWV were determined by relating the Ct values of unknown samples to an established standard curve. The standard curve was established by plotting the logarithm of seven 10-fold dilutions of a starting solution containing 21.9 ng of

plasmid DNA pCR II-TOPO (TOPO-TA cloning) with a DWV insert (from 21.9 ng to 21.9 fg), against the corresponding Ct value as the average of three repetitions. The PCR efficiency ($E = 107.5\%$) was calculated based on the slope and coefficient of correlation ($R^2$) of the standard curve, according to the following formula: $E = 10(-1/slope) - 1$ (slope $= -3.155$, $y$-intercept $= 41.84$, $R^2 = 0.999$). All primers used are shown in Supplementary Table 1.

**Impact of Clothianidin on the reproduction of *Varroa destructor*.** The artificial diet used for feeding 4th instar larvae (L4) contained D-glucose (9%), D-fructose (9%), yeast extract (2%) and royal jelly (50%)[37]. Fresh royal jelly was bought from a local supplier. Chemical analysis of royal jelly carried out by the supplier revealed no acaricides, pesticides or antibiotic contaminants. Before use, royal jelly was treated with γ-rays (25 kGy) to eliminate any possible microbial contamination.

A group of larvae received 0.01 ppm of Clothianidin-treated diet, while another group of larvae (control) received a clean diet. To prepare 100 g of Clothianidin-treated diet, 5 mg of Clothianidin were dissolved into 500 μL of acetone (solution A); then, 100 μL of solution A were diluted in 9900 μL of acetone (solution B); finally, 10 μL of solution B were dissolved in 990 μL of deionised water, which was used for the preparation of the diet.

After preparing the diet, 3–4 combs containing larvae of different ages were selected from the experimental apiary of the University of Udine, Italy. Fourth instar larvae (L4) were manually collected and transferred into sterile Petri dishes (∅ = 9 cm) containing 15 g of clean or Clothianidin-treated diet. Each Petri dish hosted 15–20 L4, for a total of 80–100 L4 per treatment per replication. Larvae maintained in Petri dishes for 24 h under controlled conditions (35 °C, 90% R.H., dark).

Mites were collected from brood cells capped in the preceding 15 h. To this aim, in the afternoon of the day preceding the experiment, when the artificial feeding of larvae was carried out, the capped brood cells of several combs were marked. The following morning, the combs were transferred to the lab and the unmarked cells, that had been capped overnight, were manually unsealed. The combs were then placed in an incubator at 35 °C and 75% R.H., where larvae and mites spontaneously emerged.

In the meantime, the larvae fed with Clothianidin (or not) that had reached the 5th instar (L5) were cleaned from the larval food and transferred into gelatin capsules (Agar Scientific ltd., ∅ = 6.5 mm) with 1 mite[38]. Infested bees were maintained in a climatic chamber under controlled conditions (35 °C, 75% R.H.) for 12 days until eclosion. From 58 to 77 L5 per experimental group per replicate were infested, for a total of 204 and 210 individuals per experimental group.

Daily, dead larvae were removed and counted. Upon eclosion, mite mortality and reproduction (i.e. fertility and fecundity) were measured by inspecting, in total, 111 and 120 mite infested honey bees fed or not with Clothianidin during the larval stage, respectively. Once separated from the infesting mite, 28 and 27 newly emerged adult bees in total, fed or not with Clothianidin during the larval stage, respectively, were stored at −80 °C for subsequent analysis aiming at assessing DWV load. The experiment was replicated 3 times.

**Modeling of *Varroa* population as affected by Clothianidin.** In order to test whether the effect of Clothianidin on *Varroa* reproduction could account for the higher mite infestation observed in colonies exposed to Clothianidin, under field conditions, we compared the data resulting from a simplified discrete time model of *Varroa* population with those obtained from the literature[13].

At each time point, our simplified discrete time model calculates *Varroa* population as follows:

- *Varroa* mites = *Varroa* mites + *Varroa* born − *Varroa* dead
- *Varroa* born = (*Varroa* mites*proportion of mites in brood cells*proportion of mites producing viable offspring)/length of reproducing phase
- *Varroa* dead = (*Varroa* mites*proportion of mites in brood cells*mortality of mites in brood cells + *Varroa* mites*(1 − proportion of mites in brood cells) *mortality of phoretic mites)/length of reproducing phase

Parameters were derived from published studies[20,39], as detailed in the Supplementary Data File. The proportion of treated mites producing viable offspring was calculated according to the results of our experiment (i.e., proportion of treated mites producing viable offspring = proportion of control mites producing viable offspring +23%). Since, the model allowed to estimate the size of *Varroa* population in treated and control colonies, whereas field studies reported the number of mites on bottom boards[13], these latter data were converted into colony infestation according to a standard coefficient derived from literature[40].

The model above was used to follow the number of mites in two experimental groups (treated and control) for the duration of the field experiment that was used as a reference. More details can be found in the Supplementary Data file.

**Statistical analysis.** The statistical tests that were used to assess significance and the relevant data are reported along the corresponding results in the Supplementary Data file. Briefly, data about melanization, encapsulation, clotting, DWV infection level, and gene expression were analyzed by means of non-parametric methods (i.e., Mann–Whitney $U$ tests in case of two samples and Kruskal–Wallis for more), the proportion of reproducing mites in different experimental groups

was tested using the Mantel–Haenszel test, clotting in adult bees exposed to different doses of Clothianidin was tested with Spearman's correlation. If necessary, probabilities were adjusted using the Bonferroni correction. Tests were performed with Excel (version 14.3.5).

**Reporting summary.** Further information on research design is available in the Nature Research Reporting Summary linked to this article.

## Data availability
Source data, including data on the immune response of bees as affected by Clothianidin treatment, reproduction of *Varroa* mites feeding on treated bees, relative expression of selected immune genes and simulations of *Varroa* population dynamics, are provided with this paper (Supplementary Data file). Primer sequences are reported in Supplementary Table 1. Source data are provided with this paper.

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

## Acknowledgements
The research leading to these results was funded by the European Union's Horizon 2020 research and innovation programme, under grant agreement No 773921 (PoshBee) and by the Italian Ministry of University, PRIN 2017 - UNICO (2017954WNT).

## Author contributions
D.A. and G.D.P. designed and performed the research, analyzed data. A.B., E.C., D.F., and A.L. performed the research. F.N. and F.P. designed the research, analyzed the data and wrote the paper. All authors revised the final version of the paper.

## Competing interests
The authors declare no competing interests.
