## [Peer Review File · Nature Communications]

Reviewers' Comments:

Reviewer #1:

Remarks to the Author:

Editors,

The manuscript entitled, "Unrecognized interactions between insecticides and parasites contribute to the decline of honey bee colonies" by Annoscia et. al describes the negative impact of an insecticide commonly used in agriculture (i.e., clothianidin) on the honey bee immune system (i.e., NF- κ B mediated melanization and encapsulation), which in turn reduces honey bee wound healing, which in turn benefits parasitic mites that feed on bees. The mite populations on clothianidin-treated bees have a greater proportion of reproducing female mites compared to controls. These findings are novel and will be of great interest to the field, and the data presented are well-supported (i.e., an excel document with all raw data was included as supporting data).

The data presented in this manuscript will also be of interest to a broader community of scientists and citizens, since honey bee populations in North American and some countries in Europe have experienced alarmingly high annual losses. Multiple factors, including pesticides and parasites/pathogens, are generally implicated in these losses, but very few studies examine the impacts of pesticides and pathogens on individual bees/bee larvae or at the mechanistic level. This work presented in this paper is an excellent example of this and it will thus be very impactful on the field.

Minor points to clarify or address before publication include:

1. Figure 1c, Figure 2 a –

It would be more accurate to label the y-axis as "DWV RNA copies per xx ng total RNA", since both DWV genomes and transcripts are detected by RT-qPCR. The authors could also just include the sample description ((i.e., copy number in 50 ng of total RNA or is it per larva?) in the figure legend (rather than the axis label), but it should be included somewhere so that the reader can interpret the copy number correctly.

It is great that the authors included the raw data as a supplemental file.

The authors could consider graphing the data for individual bees (i.e., as dots) and present the data as box-and-whisker plots, but this is an optional suggestion.

The supplemental information should include the DWV primer sequence and a brief description of the protocol so that readers do not have to go to citation #12 to find this information. The DWV abundance axis would be easier to read with labels like:

0, 10, 101, 102, 103, etc., but again this is an optional suggestion.

2. Lines 57-59. The authors should include a supplemental figure with RT-qPCR data to support their statement "that early season larva had low to undetectable levels of DWV".

While this is likely true based on the seasonality of mite infestation and DWV infections the paper would be enhanced by including this supporting data.

3. Figure 2 – The figure should include NF- κ B (labeled as Dorsal-1A) expression data (in Supplemental Figure 2) to support the model proposed in Figure 2C and/or additional text that suggests the mechanisms of NF- κ B inhibition.

4. Line 73 – The authors could include more information about the role of melanization and encapsulation in honey bee antimicrobial immune responses and add a brief description about how

these processes are connected to the NF- κ B transcription factor. As well as, how NF- κ B responses are governed by transcription (i.e., at the level of gene expression) and by binding or not to NF- κ B suppressor protein (i.e., Cactus/I κ B). This would better clarify the findings in presented in the manuscript.

5. Line 84 – mentions NF- κ B activation – which is actually depression (i.e., release from Cactus/I κ B). This could be made clearer in the text (particularly since NF- κ B expression has a trend of reduced expression in clothianidin treated bees, but the reduction was not statistically significant). Reduced (or hindered) NF- κ B signaling was also indirectly measured by quantifying the expression of one NF- κ B regulated gene, and by melanization and encapsulation assays. The authors could also examine the expression of additional NF- κ B regulated genes, but this is not required for publication.

6. Line 76 - The authors could include additional references regarding the field relevance of 0.01 ppm and 0.05 ppm clothianidin exposure to honey bee larva.

7. Lines 94-95, and Lines 119-121 - Indicate that DWV infection impairs NF- κ B activation, but do not elaborate fully – this section would be enhanced by including a brief description of their previous work (i.e., from Di Prisco et al PNAS 2015 “his result indicates that DWV infection could have a negative effect on NF- κ B activation by enhancing the transcription of its negative modulator Amel\LRRL.”).

8. Lines 101 – and 102. Please include the “symptomatic level” DWV copy number (in relation to total RNA), so that readers do not have to go to those citations in order to contextualize the results presented. It is also not easy to determine the DWV copy number in this study, since the figure legend does not include the total RNA information, which was also not clear in the methods.

9. Lines 122 – 125:

Was the proportion reproducing females (to total mite number (i.e., males and females) or in relationship to non-reproducing females only?

10. Line 139 – include a brief description of the “simple model” in the main text of the paper

11. Figures or figure legends should indicate the number of samples per biological replicate (i.e., independent experiments) and the number of biological replicates (e.g., 2?) that were carried out. Line 77 – I think indicates that two biological replicates each with 15 honey bee larva were assessed for Figure 1, as does the excel file, but this could be made clearer in the text.

12. Line 98 –Could the authors speculate in the discussion why a dose dependent effect was not observed. Again, this is not required for publication – I am just curious.

13. Lines 97, 115 – suggest changing the word “low” to “small” – although the results do not seem small or insignificant to me.

Reviewer #2:

Remarks to the Author:

This work continues a successful effort by this team to connect field relevant exposures to a

neonicotinoid pesticide with a decrease in immune function for honey bees. Unlike past work, which focused on gene expression levels, this study focusses on the abilities of bees to melanize foreign internal objects. Bees exposed to clothianidin showed reduced immune function and, in parallel or with causality, an increase in the virus Deformed wing virus. This virus is also known to impact honey bee immune function and thus might act synergistically with the pesticide to impact bee defenses, as well as bee health. Added to the mix this time is the parasitic mite *Varroa destructor*, which also might benefit from reduced immune function by honey bees since this could enable mites to feed on bee bodies more easily.

The experiments are fairly straightforward, and often carry a dose range showing dose-dependent effects, which strengthens the arguments. The effects are subtle in many of the assays, but might lead to high impacts at the colony level.

Histograms are a poor choice for exploring data visually. I would convert these all to a scattered/jittered dot display so it is possible to see how the data points align, and how responsive each bee is to the stress. This is especially important because, as the authors state (lines 119-121) the disease-inducing effects are apparently more pronounced only when virus loads are high. Seeing the actual data at the level of bees will build confidence in the generality of the results. Melanization, while a great surrogate for immune-competence, is notoriously noisy, and it would help to visualize that variation alongside the treatment effects.

I am not sure the prior references suggesting higher mite levels in colonies exposed to neonicotinoids are the best references. Dively et al. measured effects of imidacloprid not clothianidin, and their treated colonies had a range of changes that might have triggered higher mite levels, most notably higher levels of capped brood (the stage in which mites reproduce) in treated colonies. Are there no field studies showing an increase of mites with clothianidin? Alburaki et al measured colonies exposed to corn treated with thiamethoxam, and again numerous colony changes that might be correlated with mite numbers. The results from the present study also support an impact on mite numbers, but to me the strongest inferences come from the controlled experiments here showing a difference in the abilities of mites to reproduce in the lab.

Minor edits

Line 65: delete 'the' before 'direct'

Line 120: "Therefore, the immune-suppression mediated by the viral component is reasonably negligible under these experimental conditions (i.e. early in the season, with honey bee larvae bearing low levels of DWV infection)." This could be inferred/hypothesized but I think it is still speculation, right? This project did not directly measure that.

Line 145 under field condition (13), "add an 's' after condition"

Reviewer #3:

Remarks to the Author:

The author's work is well-written (aside from a few minor typos) and represents a novel, valuable contribution to the scientific community. However, one of the key conclusions is not supported by the experiments detailed in this manuscript. This work does not show that the immunosuppressive effects of Clothianidin are what boost the proliferation of *Varroa destructor* as asserted throughout. *Varroa* infertility is still a mysterious subject to the scientific community. We have not discovered, neither is there yet a suggested link between NF- κ B signaling and the observed (and rather variable) infertility of *Varroa*. While the author has shown a fascinating link between host exposure to Clothianidin and a boost in *Varroa* fertility, the author's experiments are not sufficient to claim that the impact of Clothianidin on immune suppression is the specific sub-lethal effect of this chemical that accounts for

this observed effect as there are other sub-lethal impacts. The connection is largely speculative. Unless the authors supplement this work with experiments that indicate *Varroa* fertility is suppressed by the proper functioning of the NF- κ B signaling, these connections\conclusions cannot be drawn from this study and should be referenced as speculation rather than the likely conclusions of this work.

Further, citation of the work conducted by Morfin et al. 2020 is relevant to this study and should be added to the text. These studies were likely being conducted at the same time accounting for the lack of citation in the original manuscript.

More detail would be helpful in the methods answering questions such as: what kind of fishing line was used (color & brand), how many experimental units were included in each study and treatment group and whether measurements represent averages or distinct sole-measurement values.

Further detail on these points, typos, clarity edits, and general revisions can be found in the supplementary document.

Samuel Ramsey

REVIEWER COMMENTS

Reviewer #1 (Remarks to the Author):

Editors,

The manuscript entitled, “Unrecognized interactions between insecticides and parasites contribute to the decline of honey bee colonies” by Annoscia et al. describes the negative impact of an insecticide commonly used in agriculture (i.e., clothianidin) on the honey bee immune system (i.e., NF- κ B mediated melanization and encapsulation), which in turn reduces honey bee wound healing, which in turn benefits parasitic mites that feed on bees. The mite populations on clothianidin-treated bees have a greater proportion of reproducing female mites compared to controls. These findings are novel and will be of great interest to the field, and the data presented are well-supported (i.e., an excel document with all raw data was included as supporting data).

The data presented in this manuscript will also be of interest to a broader community of scientists and citizens, since honey bee populations in North American and some countries in Europe have experienced alarmingly high annual losses. Multiple factors, including pesticides and parasites/pathogens, are generally implicated in these losses, but very few studies examine the impacts of pesticides and pathogens on individual bees/bee larvae or at the mechanistic level. This work presented in this paper is an excellent example of this and it will thus be very impactful on the field.

We thank reviewer 1 for her/his recognition of our effort to characterize at mechanistic level the interactions among different stress factors that can impact honey bee health: a subject which is really worth of attention for the great implications it has both for beekeeping and crop production.

Minor points to clarify or address before publication include:

1. Figure 1c, Figure 2 a –

It would be more accurate to label the y-axis as “DWV RNA copies per xx ng total RNA”, since both DWV genomes and transcripts are detected by RT-qPCR. The authors could also just include the sample description ((i.e., copy number in 50 ng of total RNA or is it per larva?) in the figure legend (rather than the axis label), but it should be included somewhere so that the reader can interpret the copy number correctly.

Thank you for spotting this incomplete information. We changed the y-axis labels from “DWV genome copies” to “DWV RNA copies per bee” and revised figure legends as suggested by the reviewer.

It is great that the authors included the raw data as a supplemental file.

Thank you for noting this point: we believe that this is the most transparent option, allowing the reader to check personally whatever detail of data and statistical analysis. Actually, this is why we omitted some details, such as sample size and number of replicates, but we are happy to include this information in the text, as requested by the reviewers, because it will certainly help the reader.

The authors could consider graphing the data for individual bees (i.e., as dots) and present the data as box-and-whisker plots, but this is an optional suggestion.

We replaced all the figures regarding viral loads from bars to scattered/jittered dot display. Histograms regarding melanization and encapsulation were converted to whisker plots.

The supplemental information should include the DWV primer sequence and a brief description of the protocol so that readers do not have to go to citation #12 to find this information. The DWV abundance axis would be easier to read with labels like:

0, 10, 101, 102, 103, etc., but again this is an optional suggestion.

We added a short description of the protocol in the supplementary materials and methods and the primers for DWV analysis in supplementary table 1.

We hope that the revised figures are now easier to read.

2. Lines 57-59. The authors should include a supplemental figure with RT-qPCR data to support their statement “that early season larva had low to undetectable levels of DWV”. While this is likely true based on the seasonality of mite infestation and DWV infections the paper would be enhanced by including this supporting data.

DWV levels in the bees used for the implantation experiments and the Varroa rearing experiments are reported in figures 1c, 1f and 2a, respectively. As we commented in lines 116-124 and 135-141, the observed levels (i.e. 10^6) are really low compared to those observed later in the season, when DWV level can reach 10^{15} - 10^{18} copies per bee (as widely reported in the literature we cite), associated with a severe immune depression.

3. Figure 2 – The figure should include NF- κ B (labeled as Dorsal-1A) expression data (in Supplemental Figure 2) to support the model proposed in Figure 2C and/or additional text that suggests the mechanisms of NF- κ B inhibition.

*The mechanism underlying the negative effect of Clothianidin on NF- κ B has been described by us in a previous paper (Di Prisco et al., Proc. Natl. Acad. Sci. USA **110**, 18466–18471, 2013), which is now mentioned in the figure legend, along with the data that confirm this effect in the current study.*

4. Line 73 – The authors could include more information about the role of melanization and encapsulation in honey bee antimicrobial immune responses and add a brief description about how these processes are connected to the NF- κ B transcription factor. As well as, how NF- κ B responses are governed by transcription (i.e., at the level of gene expression) and by binding or not to NF- κ B suppressor protein (i.e., Cactus/I κ B). This would better clarify the findings in presented in the manuscript.

We added a paragraph that explains the rationale behind the experimental approach used, describing why the immune response possibly affects the host nutritional suitability for the feeding mite and the central role of NF- κ B and of any stress factor that interferes with its signaling activity (see lines 78-89).

5. Line 84 – mentions NF- κ B activation – which is actually depression (i.e., release from Cactus/I κ B). This could be made clearer in the text (particularly since NF- κ B expression has a trend of reduced expression in clothianidin treated bees, but the reduction was not statistically significant). Reduced (or hindered) NF- κ B signaling was also indirectly measured by quantifying the expression of one NF- κ B regulated gene, and by melanization and encapsulation assays. The authors could also examine the expression of additional NF- κ B regulated genes, but this is not required for publication.

We have better described the underlying mechanistic aspects accounting for the reported results and explained why we focused on Amel/102 -a gene under NF- κ B control- given its importance in melanization and encapsulation response, which is relevant in the physiological responses we measured in this work (see lines 101-111).

We have already published data demonstrating that Clothianidin negatively affects the transcription of AMP genes (e.g. Apidaecin), and have added this information in the text by mentioning the original papers (4, 12).

6. Line 76 - The authors could include additional references regarding the field relevance of 0.01 ppm and 0.05 ppm clothianidin exposure to honey bee larva.

We added 3 more references to support our statement (see line 158).

7. Lines 94-95, and Lines 119-121 - Indicate that DWV infection impairs NF- κ B activation, but do not elaborate fully – this section would be enhanced by including a brief description of their previous work (i.e., from Di Prisco et al PNAS 2015 “his result indicates that DWV infection could have a negative effect on NF- κ B activation by enhancing the transcription of its negative modulator Amel/LRR/.”).

We have better explained the underlying mechanistic aspects, by adding/rewriting some paragraphs (see lines 112-113).

8. Lines 101 – and 102. Please include the “symptomatic level” DWV copy number (in relation to total RNA), so that readers do not have to go to those citations in order to contextualize the results presented. It is also not easy to determine the DWV copy number in this study, since the figure legend does not include the total RNA information, which was also not clear in the methods.

We added a quantitative figure for the “symptomatic level” of DWV copy number and a reference (see lines 120-122).

We have explicitly indicated that the data reported in the graphs are the number of DWV RNA copies per bee.

9. Lines 122 – 125:

Was the proportion reproducing females (to total mite number (i.e., males and females) or in relationship to non-reproducing females only?

The fertility of Varroa has been the focus of extensive research in the past. It represents the proportion of mites which reproduced (i.e. produced at least one offspring) out of the total mites entering a brood cell for reproduction (or used for the infestation of artificial rearing cells, as in our case). We added a few more words to better explain this concept and cited a paper where we extensively deal with this aspect of the mite’s life cycle (see lines 142-143).

10. Line 139 – include a brief description of the “simple model” in the main text of the paper

We added the following sentence at the end of the paragraph (see lines 164-166):

“Briefly, in our discrete time model, the mite population is calculated on a daily basis, using standard parameters as derived from the literature and corrected to include the observed effect of Clothianidin on mite reproduction.”

11. Figures or figure legends should indicate the number of samples per biological replicate (i.e., independent experiments) and the number of biological replicates (e.g., 2?) that were carried out.

We added this information in the figure legends.

Line 77 – I think indicates that two biological replicates each with 15 honey bee larva were assessed for Figure 1, as does the excel file, but this could be made clearer in the text.

Correct. To better clarify, we have now added the sample size in each figure legend.

12. Line 98 –Could the authors speculate in the discussion why a dose dependent effect was not observed. Again, this is not required for publication – I am just curious.

We believe that the peculiar viral dynamics, involving thresholds for entering the exponential viral replication, makes the interpretation of the lack of a dose-response in case of two treatments differing by a factor 5 particularly challenging.

However, a dose-dependent effect was observed on adult bees, when a broader range of experimental doses was used.

Therefore, to take into account the reviewer's comment, we added the following sentence (lines 93-95):

“Moreover, parallel studies on adult honey bees showed that the observed effects are not stage specific and are clearly dose-dependent within a broader range of sub-lethal doses”.

13. Lines 97, 115 – suggest changing the word “low” to “small” – although the results do not seem small or insignificant to me.

We changed “low” with “small”.

We agree with reviewer that the change may not seem that small in general but it can be regarded as small if compared with the change that can be observed under different conditions (e.g. when a mite is added to a L5 larva and the viral load can jump from 10^6 to 10^{15} ; see ref. 20; or when the basal level of viral load in controls is higher than in this study and the resulting relative increase can be remarkably pronounced; see ref. 4 and 12), that can lead to values having a dramatic impact on immune-competence.

Reviewer #2 (Remarks to the Author):

This work continues a successful effort by this team to connect field relevant exposures to a neonicotinoid pesticide with a decrease in immune function for honey bees. Unlike past work, which focused on gene expression levels, this study focusses on the abilities of bees to melanize foreign internal objects. Bees exposed to clothianidin showed reduced immune function and, in parallel or with causality, an increase in the virus Deformed wing virus. This virus is also known to impact honey bee immune function and thus might act synergistically with the pesticide to impact bee defenses, as well as bee health. Added to the mix this time is the parasitic mite Varroa destructor, which also might benefit from reduced immune function by honey bees since this could enable mites to feed on bee bodies more easily.

The experiments are fairly straightforward, and often carry a dose range showing dose-dependent effects, which strengthens the arguments. The effects are subtle in many of the assays, but might lead to high impacts at the colony level.

We thank reviewer 2 for his/her positive and in-depth comments, which reflect at the best the content of our work.

Histograms are a poor choice for exploring data visually. I would convert these all to a scattered/jittered dot display so it is possible to see how the data points align, and how responsive each bee is to the stress. This is especially important because, as the authors state (lines 119-121) the disease-inducing effects are apparently more pronounced only when virus loads are high. Seeing the actual data at the level of bees will build confidence in the generality of the results. Melanization, while a great surrogate for immune-competence, is notoriously noisy, and it would help to visualize that variation alongside the treatment effects.

We replaced all the figures as requested. See our reply to the same comment by reviewer 1.

I am not sure the prior references suggesting higher mite levels in colonies exposed to neonicotinoids are the best references. Dively et al. measured effects of imidacloprid not clothianidin, and their treated colonies had a range of changes that might have triggered higher mite levels, most notably higher levels of capped brood (the stage in which mites reproduce) in treated colonies. Are there no field studies showing an increase of mites with clothianidin? Alburaki et al measured colonies exposed to corn treated with thiamethoxam, and again numerous colony changes that might be correlated with mite numbers. The results from the present study also support an impact on mite numbers, but to me the strongest inferences come from the controlled experiments here showing a difference in the abilities of mites to reproduce in the lab.

We agree that higher mite populations reported under field conditions after neonicotinoid treatment may be due to other additional factors, and that many studies available in the literature refer to different neonicotinoid insecticides. However, it is worth noting that Clothianidin is a common metabolite of Thiametoxam (see: Fan, Y. & Shi, X. Characterization of the metabolic transformation of thiamethoxam to clothianidin in Helicoverpa armigera larvae by SPE combined UPLC-MS/MS and its relationship with the toxicity of thiamethoxam to Helicoverpa armigera larvae. J Chromatogr B. 1061, 349–355 (2017)) and, in fact, Alburaki et al., did find Clothianidin in corn flowers from Thiametoxam treated fields.

In any case, we do not intend to use previous field data as a proof of what we propose here. We only say that our results could explain, at least in part, the unexpected proliferation of mites in relation to neonicotinoid treatments.

Minor edits

Line 65: delete ‘the’ before ‘direct’

Done.

Line 120: “Therefore, the immune-suppression mediated by the viral component is reasonably negligible under these experimental conditions (i.e. early in the season, with honey bee larvae bearing low levels of DWV infection).” This could be inferred/hypothesized but I think it is still speculation, right? This project did not directly measure that.

Our statement on negligible effects on immune competence by the recorded small levels of DWV titer is based on evidence available in the literature, which is now cited in the context of a sentence that was changed as follows (lines 138-141):

“Based on previous data (20), we can assume that the immune-suppression mediated by the viral component is reasonably negligible under these experimental conditions (i.e. early in the season, with honey bee larvae bearing low levels of DWV infection).”

Line 145 under field condition (13), “add an ‘s’ after condition”

Done.

Reviewer #3 (Remarks to the Author):

The author's work is well-written (aside from a few minor typos) and represents a novel, valuable contribution to the scientific community. However, one of the key conclusions is not supported by the experiments detailed in this manuscript. This work does not show that the immunosuppressive effects of Clothianidin are what boost the proliferation of Varroa destructor as asserted throughout. Varroa infertility is still a mysterious subject to the scientific community. We have not discovered, neither is there yet a suggested link between NF- κ B signaling and the observed (and rather variable) infertility of Varroa. While the author has shown a fascinating link between host exposure to Clothianidin and a boost in Varroa fertility, the author's experiments are not sufficient to claim that the impact of Clothianidin on immune suppression is the specific sub-lethal effect of this chemical that accounts for this observed effect as there are other sub-lethal impacts. The connection is largely speculative. Unless the authors supplement this work with experiments that indicate Varroa fertility is suppressed by the proper functioning of the NF- κ B signaling, these connections/conclusions cannot be drawn from this study and should be referenced as speculation rather than the likely conclusions of this work.

Reviewer 3 is right when he says that Varroa infertility “is still a mysterious subject”; however, it is worth noting that a great deal of work on the subject has been carried out in the past providing important information that are relevant for the issue discussed here. On the ground of those old studies (for a review, see ref. 9 and references therein), we can assume that varroa fertility is strongly influenced by several factors, including: the conditions of the bee larva (e.g. fertility is drastically reduced if a reproducing mite is placed on a bee larva after the first 24 hours post cell sealing), the physiological conditions of the mite (e.g. fertility is influenced by the duration of the phoretic phase), the conditions of the rearing environment (e.g. fertility is drastically reduced in unsuitable cells even on a suitable host).

We are well aware of this, and by no mean we would like to state that mite fertility depends only upon NF- κ B signaling, which, per se, would be a misleading statement. What we mean here is that, under optimal conditions (i.e. a mite in the right physiological state, on a suitable host, in a suitable cell), the parasite needs to feed sufficiently to support oocyte development and subsequent oviposition, and this can be affected by the persistence of a previous feeding hole, which can be limited by immune reactions under NF- κ B control or promoted by any immunosuppressive factor, such as Clothianidin or DWV infection. All mechanistic details involved are grounded on experimental data. The really lacking evidence is a measure of the food uptake and of the induced metabolic changes associated with different feeding efficiencies generated by the exposure to Clothianidin. This is not trivial to do and will be the focus of future studies.

Obviously, there are many different experimental approaches that can be used to lend further support to this fascinating hypothesis, and the reviewer provides some interesting suggestions. However, the “suppression of Varroa fertility by proper functioning of NF- κ B”, as a suggested experimental evidence to provide support to our results, is conceptually difficult to understand, because the proper functioning of NF- κ B is the basal condition occurring in controls of our experiments, which are the regular host on which the mite develops and reproduces, even though with a fertility lower than that observed on Clothianidin treated experimental honey bee larvae. We may miss something in the reasoning behind this suggestion, which could be thoroughly explored in future work, building upon what reported in this manuscript and in our previous publications. The body of experimental evidence provided here and the importance of the results obtained so far have suggested the preparation of the present manuscript which, in our opinion, is worth to share

with the scientific community as it stands. Future work will certainly explore some of the multiple hypotheses provided by the reviewer.

Reviewer 3 is correct when he notes that we do not experimentally show that the increased feeding promotes fertility and Clothianidin may trigger increased fertility through another unrecognized mechanism.

For this reason, to tone down our reasonable conclusion we have changed the sentence as follows

From:
“The observed increase of the mite’s fertility is likely due to the fact that its feeding activity is facilitated by Clothianidin induced immune-suppression of the host.”

To (lines 148-150):

“The observed increase of fertility in mites parasitizing Clothianidin treated bee larvae is possibly due to the fact that their feeding activity is facilitated by the induced immune-suppression of the host.”

Hope the new wording is sufficient to make clear this point.

Further, citation of the work conducted by Morfin et al. 2020 is relevant to this study and should be added to the text. These studies were likely being conducted at the same time accounting for the lack of citation in the original manuscript.

Morfin et al. 2020, that was not yet published at the time we submitted our manuscript, has now been cited and included in the reference list.

More detail would be helpful in the methods answering questions such as: what kind of fishing line was used (color & brand), how many experimental units were included in each study and treatment group and whether measurements represent averages or distinct sole-measurement values.

We have added more details in the supplementary materials and methods about the fishing line used in the implantation experiments.

Experimental units and other details: we checked the manuscript throughout once more to make sure that those details are not missing. Following the suggestion by reviewer 1, we now indicate the number of samples per replicate in the figure legends.

Further detail on these points, typos, clarity edits, and general revisions can be found in the supplementary document.

Below is a list of the points raised in the supplementary document by reviewer 3 and our answer to them.

Line 26

This work does not show that the immunosuppressive effects of Clothianidin are what boost the proliferation of Varroa destructor. This work does show that Clothianidin has a direct impact on some elements of the immune response and that treatments of Varroa fed on immunocompromised bees had a higher rate of fertility but the connection between the two is largely speculative and should be stated as such. Unless you supplement this work with experiments that indicate Varroa reproduction is suppressed by the proper functioning of the NF-kB signaling, these proposed connections should be stated as speculative or removed.

This is the same concept expressed above, in the general comments provided by the reviewer. We agree with reviewer 3 that we do not directly demonstrate that immune-suppression enhances mite feeding and reproduction. In principle, there could be some other unknown mechanisms accounting

for what observed. However the fact that the immunosuppression differently induced by DWV similarly enhances the mite's fitness (see ref. 4) lends indirect support to our interpretation, indicating that the reduced defense barriers are the common element shared by Clothianidin exposure and DWV infection, whatever is the upstream mechanism of induction. Based on this, our hypothesis (i.e. enhanced feeding efficiency) seems to us the most reasonable. Whatever the mechanism, the experimental evidence clearly shows that the insecticide-induced immunosuppression determines a significant increase of mite's fertility. Nevertheless, to account for the lack of a direct proof between enhanced feeding and immunosuppression, we changed the sentence as follows (lines 29-31): "Here we show that the immune-suppression exerted by Clothianidin is associated with an enhanced fertility of the parasitic mite Varroa destructor, as a possible consequence of a higher feeding efficiency."

Line 31

Sentence Structure/Clarity:

This sentence should be revised to read more clearly. Example provided below:

"Neonicotinoids entered the pesticide market in 1990 rapidly becoming the most widely used insecticide molecules worldwide."

We changed the text as suggested by reviewer 3 (see lines 34-35).

Line 38

Your citations in this paper should include mention of: Morfin et al. 2020's work on the interaction between Varroa destructor and sublethal impacts levels of clothianidin on larval honey bee immunity and differential gene expression. This work was likely not available to you when you first submitted as it was published recently.

At the time of submission the paper by Morfin et al. was not available. We are happy to cite this paper (see comment below on line 38).

Line 40

Typo: preminent

Corrected.

Line 38

Revision: Citation to Morfin work should be added here as well.

We added a citation to Morfin et al. where we spoke about the "synergistic negative effect of the association between Varroa and neonicotinoid insecticides on honey bees" (see lines 50-52).

Line 62

Revise:

For the sake of reproducibility, you need to define your standard for "low titers of DWV" and was this standard measured or simply assumed because of the time of year

We reported in the article the DWV levels found in the bees used for our experiments (see figures 1 and 2).

Line 103

Define: "immune disguise"

We changed immune disguise to immune depression.

Line 109

The most up to date data suggests that Varroa are interested in the uptake of fat body tissue rather than hemolymph when feeding on both adult bees and brood.

In our opinion, the data reported in the most recent paper on the subject (Ramsey et al. on PNAS January 29, 2019, 116 (5) 1792-1801) are not sufficient to disregard/ignore a vast, consistent and consolidated body of knowledge supporting the view that the mite feeds on bee's haemolymph, which, of course, may well contain materials/tissue debris coming from the fat body. Because there is no need in this paper to enter this debate, we prefer to replace "haemolymph uptake" with "food uptake".

Line 110

Key Revision:

Your work does not distinguish precisely how clothianidin boosts reproduction in Varroa. The relationship between the immune response and Varroa here is assumed. In such cases, that assumption should be clearly stated and other potential explanatory measured should be detailed. An example would be hormesis which would account for the heightened reproductive rate in the parasitic mites which would also be exposed to sublethal levels of this insecticide in the larvae that serve as hosts in these treatments. Further, fat body tissue and hemocytes have been shown to mechanically close wounds in fruitflies working in tandem with the melanization process with the implication that they do the same in other insects (Franz et al. 2018). Sub-lethal levels of Clothianidin have been shown to increase hemocyte counts (Morfin et al. 2020) which would be expected to accelerate the process of wound closure as larger wounds resulted in the mobilization of more hemocytes and fat body cells to close the found more quickly. The conclusion that Varroa feeding has better access to a pervious feeding site does not follow from the work here as it fails to show that wound closure is slowed or halted by Clothianidin exposure. While melanization is diminished the promotion of more hemocytes by Clothianidin appears to promote a response that would potentially promote swifter closing of the feeding wound.

For the reasons explained above, we admit that the causal link connecting immune-depression, enhanced/improved feeding and reproduction has not been directly demonstrated experimentally. For these reasons, throughout the manuscript, we have toned down the claims when referring to this very plausible hypothesis, which remains very well defined but clearly stated as such. In our opinion, mentioning in the text a series of hypothetical explanations, by far less likely than the one we propose, could be disorienting for the reader, who could get the impression that nothing can be concluded based on the presented experimental data. This is absolutely not the case, as it clearly appears also from the positive opinions expressed by the other two reviewers. Any scientific paper will always leave some open questions, the point is to judge if what communicated is considered worth to be shared with the scientific community. We are genuinely convinced that this manuscript, without overstating any claim, can be a good addition to the scientific literature,

Line 112

You are corroborating the hypothesis that "Varroa reproductive rate is impacted by host exposure to clothianidin" but not how clothianidin does so. As a result, your earlier assertions should be referenced as speculative.

Please see comments above and related changes in the text to tone down our claims.

Line 127

Revision for Clarity:

The observed increase was observed in the specific treatment population and not an individual mite. To say "the mite's fertility" may be confusing. Please revise

We changed the sentence:

"The observed increase of the mite's fertility is likely due to the fact that its feeding activity is facilitated by Clothianidin induced immune-suppression of the host."

To (lines 148-150):

"The observed increase of fertility in mites parasitizing Clothianidin treated bee larvae is possibly due to the fact that their feeding activity is facilitated by the induced immune-suppression of the host."

Line 128

Key Revision:

Without experiments aimed at determining the reason for this increase in fertility with clothianidin treatment, these sorts of statements are not appropriate. Without further data targeted at the mechanisms driving this process, this assertion should be stated as a "potential" explanation not a "likely" explanation. The reasons for Varroa infertility are still mysterious to the scientific community. It is not clear that the infertility is linked to the honey bee's immune response or how Clothianidin would impact a process of which we we don't yet understand the underpinnings.

We changed "likely" with "possibly".

Moreover, it is worth to consider that in our paper we do not talk about infertility but enhanced fertility compared to regular fertility of controls, with proper functioning of NF-kB signaling and of immune barriers.

Line 134

Revise:

The word here should be "comparable"

Corrected.

Line 135

Clarity Edit:

Nearby would be better than "by". Otherwise this sentence could read as the treated crops main the bees.

Corrected.

Figure 2

These abbreviations make your variables difficult to distinguish. You appear to have enough space to write out the words "control" and "clothianidin"

We now use the complete words.

Fig. 2

Methods Revision:

In line 122 of the methods (the supplementary file), the word "length" is spelled incorrectly. Please go through these manuscripts thoroughly to ensure all typos are found and corrected.

Corrected. We also checked the manuscript for more possible typos.

Fig. 2

How many experimental units were used in this study?

We have now added this information in the legend.

Reviewers' Comments:

Reviewer #1:

Remarks to the Author:

This manuscript has been improved by the review process and will be an important contribution to the field.

It seems the authors may have misinterpreted the request to label the y-axis of Figures 1c and Figure 2 as "DWV RNA copies per xx ng RNA", NOT on a per bee (or per larva) basis since the total amount of RNA extracted from a bee or larva varies with each sample. Furthermore, RT-qPCR estimates the starting quantity of cDNA copies in each well, which directly relates to the amount of RNA in the RT reaction (since cDNA is not usually quantified after RT reactions). Therefore the axis should be "DWV RNA copies per xx ng RNA", (either the amount of RNA in each reaction, or they could do additional calculations to for "total RNA" per bee, but this would not be ideal).

Again, since the amount of RNA obtained for each bee sample was not reported, it is important to change the axis level, so that readers can get a relative sense of virus abundance without having to consider the efficiency of bee sample RNA extraction (i.e., what was the total RNA obtained for each sample).

Reviewer #2:

Remarks to the Author:

This manuscript is much improved and the data remain robust with the new analyses. It is still unclear whether the same pathways that limit melanization of foreign objects placed inside bees are linked with those effecting parasite reproduction, but that connection will have to be made using bee-by-bee experiments perhaps. It is plausible and this paper makes the best case to date for such an interaction.

The overall message that a physiological; change induced by clothianidin leads ot lower immunocompetence and higher virus levels is strong, and that is an important connection.

Reviewer #3:

Remarks to the Author:

The authors detailed responses and subsequent edits were satisfactory. The work was already of sufficient scientific merit in its findings and robust methods to warrant publication without expanding the scope of its conclusions beyond the scope of the experiments.

REVIEWERS' COMMENTS

Reviewer #1 (Remarks to the Author):

This manuscript has been improved by the review process and will be an important contribution to the field.

It seems the authors may have misinterpreted the request to label the y-axis of Figures 1c and Figure 2 as “DWV RNA copies per xx ng RNA”, NOT on a per bee (or per larva) basis since the total amount of RNA extracted from a bee or larva varies with each sample. Furthermore, RT-qPCR estimates the starting quantity of cDNA copies in each well, which directly relates to the amount of RNA in the RT reaction (since cDNA is not usually quantified after RT reactions). Therefore the axis should be “DWV RNA copies per xx ng RNA”, (either the amount of RNA in each reaction, or they could do additional calculations to for "total RNA" per bee, but this would not be ideal).

Again, since the amount of RNA obtained for each bee sample was not reported, it is important to change the axis level, so that readers can get a relative sense of virus abundance without having to consider the efficiency of bee sample RNA extraction (i.e., what was the total RNA obtained for each sample).

We fully understand the issue raised by the Reviewer and agree that reporting the number of DWV genome copies per ng of RNA is the most appropriate and direct way to represent the abundance of the virus. Indeed, the estimate of viral load per experimental individual requires the additional calculation of the total RNA extracted, as we did, which is also influenced by the extraction efficiency. Notwithstanding the reasonable comment made by the reviewer, we would prefer to maintain the original label on the Y axis of the mentioned figures because referring to the viral load per bee or larva is of great help for the reader as it allows a direct comparison of our data with data available in the literature that are mentioned in the manuscript.

Reviewer #2 (Remarks to the Author):

This manuscript is much improved and the data remain robust with the new analyses. It is still unclear whether the same pathways that limit melanization of foreign objects placed inside bees are linked with those effecting parasite reproduction, but that connection will have to be made using bee-by-bee experiments perhaps. It is plausible and this paper makes the best case to date for such an interaction.

The overall message that a physiological; change induced by clothianidin leads to lower immunocompetence and higher virus levels is strong, and that is an important connection.

We appreciate this positive comment and fully agree that further experimental work is required to directly demonstrate the functional link between reduced immunocompetence and *Varroa* proliferation. The possible avenues of research in this direction were tentatively outlined in our previous rebuttal letter.

Reviewer #3 (Remarks to the Author):

The authors detailed responses and subsequent edits were satisfactory. The work was already of sufficient scientific merit in its findings and robust methods to warrant publication without expanding the scope of its conclusions beyond the scope of the experiments.

We appreciate the positive assessment by the Reviewer of the revised version of the manuscript.